# General practitioners' consultation counts and associated factors in Swiss primary care – A retrospective observational study

**Yael Rachamin**[ID]\*, **Rahel Meier, Thomas Grischott**[ID]**, Thomas Rosemann, Stefan Markun**

Institute of Primary Care, University and University Hospital Zurich, Zurich, Switzerland

\* yael.rachamin@usz.ch

## Abstract

### Background

Research on individual general practitioner (GP) workload, e.g. in terms of consultation counts, is scarce. Accurate measures are desirable because GPs' consultation counts might be related to their work satisfaction and arguably, there is a limit to the number of consultations a GP can hold per day without jeopardizing quality of care. Moreover, understanding the association of consultation counts with GP characteristics is crucial given current trends in general practice, such as the increasing proportion of female GPs, part-time work and group practices.

### Aim

The aim of this study was to describe GPs' consultation counts and efficiency and to assess associations with GP and practice variables.

### Methods

In this retrospective observational study we used routine data in electronic medical records obtained from 245 Swiss GPs in 2018. We described GPs' daily consultation counts as well as their efficiencies (i.e. total consultation counts adjusted for part-time work) and used hierarchical linear models to find associations of the GPs' total consultation counts in 2018 with GP- and practice-level variables.

### Results

The median daily consultation count was 28 over all GPs and 33 for full-time working GPs. Total consultation counts increased non-linearly with part-time status, with high part-time working GPs (60%-90% of full-time) being equally or more efficient than full-time workers. Excluding part-time status in the regression resulted in higher consultation counts for male GPs working in single practices and with older patients, whereas part-time adjusted consultation counts were unaffected by GP gender and practice type.

**Data Availability Statement:** Legal restrictions in Switzerland prohibit public release of original patient data without consent. The authors' fully anonymized data is exempt from these legal

restrictions. However, the data could be de-anonymized by individuals or organizations such as health insurers who have overlapping data (e.g. patient date of birth and consultation dates). Data access queries can be sent to Rahel Meier (rahel.meier@usz.ch) after clearance by the local ethics committee, or to the Kantonale Ethikkommission Zürich (Local Ethics Committee of the Canton of Zurich) (Info.KEK@kek.zh.ch).

**Funding:** The authors received no specific funding for this work.

**Competing interests:** The authors have declared that no competing interests exist.

## Conclusion

Female gender, part-time work in the range of 60%-90% of full-time, and working in group practices do not decrease GP efficiency. However, the challenge of recruiting sufficient numbers of GPs remains.

## Introduction

Research on individual general practitioner (GP) workload, e.g. in terms of consultation counts, is scarce. So far, studies of consultation counts have depended on self-reports which are highly susceptible to recall bias and disregard day-to-day variability and often also part-time status [1–3]. More accurate measures are desirable because GPs' consultation counts might be related to their work satisfaction [4] and arguably, there is a limit to the number of consultations a GP can hold per day without jeopardizing quality of care.

Moreover, current trends suggest an increasing demand for primary care consultations [5–7]. The demand is likely to increase further because of accumulating chronic conditions in the aging population [8–12]. At the same time, the GP population is changing. In Switzerland and other occidental countries, many GPs will reach retirement age soon [3, 13]. The next generation of GPs will consist of an increased proportion of female individuals and will prefer working part-time and in urban group practices [1, 3, 14–17]. Whether these next-generation GPs with different preferences will be able to fill the gap of the retiring ones is uncertain as, until now, little is known about the associations between GPs' personal characteristics and their consultation output.

Therefore, the aim of this study was to describe GPs' consultation counts and efficiency and to assess associations with GP and practice variables using electronic medical records (EMR) data.

## Methods

### 2.1 Design and setting

We conducted a retrospective observational study with data obtained from the FIRE (Family Medicine ICPC-Research using Electronic Medical Records) database. The FIRE database collects anonymized routine data exported from EMR of participating GPs from the German speaking part of Switzerland. Since the project started in 2009, 524 GPs (roughly 10% of all GPs working in the German speaking area [18]) have joined. The database holds records of over 623'000 patients and more than 6.9 million consultations (as of April 04, 2019). For this cross-sectional study, we restricted our analyses to consultations on workdays (see definition below) in 2018. According to the Ethics Committee of the Canton of Zurich, the project does not fall under the scope of the Federal Act on Research involving Human Beings (Human Research Act) [19] and therefore no ethical consent was necessary (BASEC-Nr: Req-2017-00797).

### 2.2 Participants

We included all GPs in the FIRE database with a) known age and part-time status (% full-time equivalent) in 2018. We excluded those who b) exported data of less than 10 months in 2018, c) were associated with multiple GP practices (e.g. because of changing their workplace from one practice to another), d) exported data as a group of GPs (precluding analyses of individual

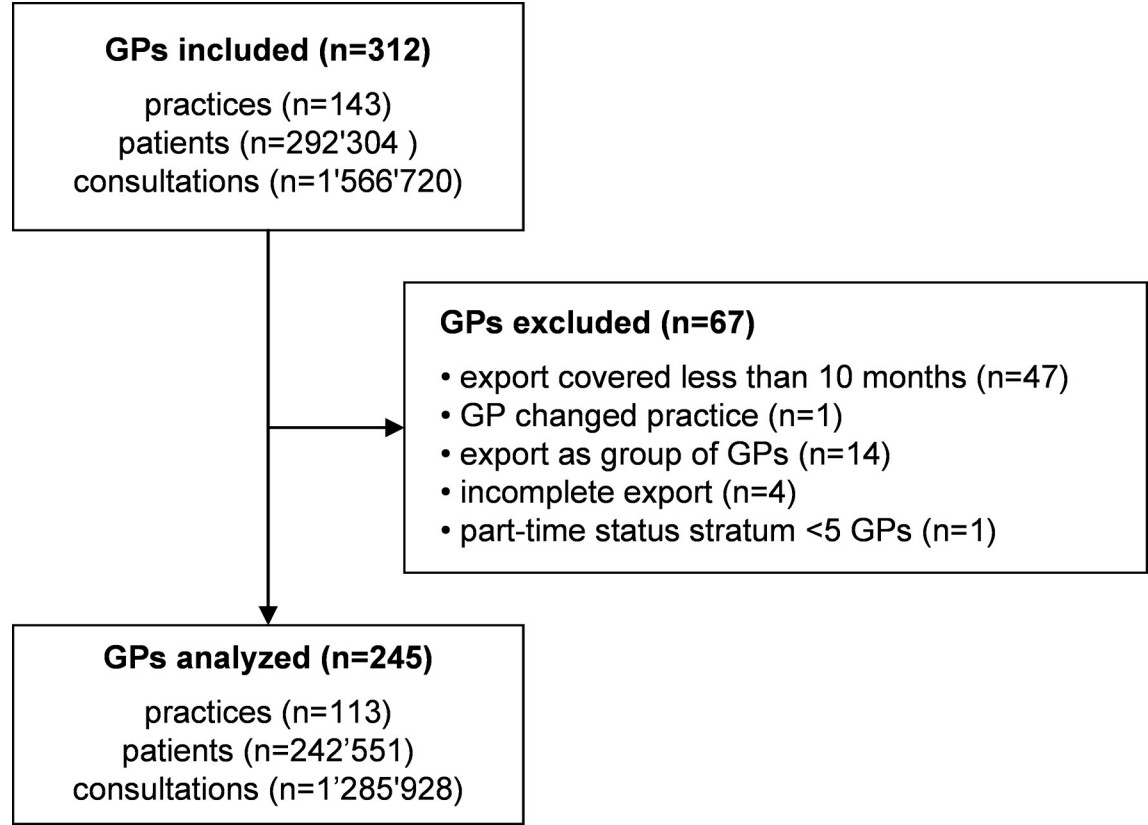

**Fig 1. Flowchart.**

GPs' consultation counts), e) showed evidence for false-negative consultation data (incomplete datasets) and f) belonged to part-time working strata containing < 5 GPs (disallowing meaningful summary statistics).

To improve the accuracy of the inclusion criterion and data validity, we updated the part-time status by email inquiry to all participating practices. In this way, we achieved an inclusion rate of 88% (312 of 353) of individual FIRE GPs in 2018. To improve the accuracy of the exclusion criteria, we examined the plausibility of consultation counts by manually investigating outlier GPs, defined as those with part-time adjusted total consultation counts in 2018 below the 10th or above the 90th percentile. Outlier GPs were investigated by searching their practice websites and additional internal information for other GPs that exported data under the same GP identifier (leading to exclusion under criterion d) and by checking whether they did not export properly (leading to exclusion under criterion e). The selection process is visualized in the study flowchart (Fig 1).

## 2.3 Database query, variables and definitions

From the database, we extracted consultation data for all workdays in 2018. As workdays we considered all days of the year except weekends and public holidays. Consultation data included patient information (patient identification number, age, and gender), GP information (GP identification number, part-time status, age, gender, employment contract) and practice information (practice identification number, zip code, practice type). Urbanity of the practice was determined from the zip code [20].

## 2.4 Outcomes

Outcomes of the study were:

1. GPs' daily consultation counts, stratified by part-time status

2. GPs' efficiencies (consultation counts per full-time equivalent)

3. Associations of GPs' total consultation counts in 2018 with GP-level and practice-level covariates

## 2.5 Data analysis

We described categorical data by counts and/or proportions (n, %) as appropriate and numerical data by median and interquartile range (IQR). We aggregated data to represent consultation counts for every GP and workday in 2018. From that, we determined the mean daily consultation count for every GP considering only days when the individual GP held at least one consultation. GPs' efficiencies, i.e. consultation counts per full-time equivalent, were calculated from the GPs' total consultation counts in 2018 and their part-time status. They were reported as relative efficiencies with respect to the median number of consultations of full-time working GPs in 2018.

We used hierarchical linear models with random practice effects to find associations of the GPs' total 2018 consultation counts with GP-level covariates (part-time status, gender, age group, employment contract, and consultation patient characteristics; the latter meaning median age of patients in consultations and proportion of consultations with female patients) and practice-level covariates (practice type, urbanity). Firstly, every variable was included as the only fixed effect amongst the random practice effects (crude model). Secondly, we adjusted using two multivariable models: a model adjusting for all variables except for part-time status, and a fully adjusted model. The rationale behind this was the assumption that variables potentially affect GP consultation counts both indirectly through their effects on part-time status and directly. Disregarding part-time status can therefore give hints on total effects, whereas adjusting for it blocks mediation, thus revealing direct effects. Significance was determined at the 5% level; 95% confidence intervals (CI) were reported accordingly. Variables were adapted for analysis as appropriate, i.e. GP age was categorized into age groups of 10 years and part-time status was rounded to the smaller tens digit. Consultation patient characteristics were calculated for each GP from all their consultations (thus allowing to count individual patients multiple times in order to reflect the GP perspective on consultations) and mean centered. All analyses were conducted using the R statistical package version 3.5.0 [21].

# Results

## Population

Data covered 1'285'928 consultations with 242'551 individual patients and 245 GPs in 113 practices. Of the GPs, 38.0% were female and the GPs' median age was 51 (IQR = 43 to 58). The majority worked in group practices (87.3%), was self-employed (64.7%) and located in urban areas (75.5%). On average (median), the GPs worked on 88.8% (IQR = 75.5% to 97.6%) of all workdays and held 4'843 (IQR = 3'318 to 6'908) consultations in 2018 with 1'125 (IQR = 836 to 1'477) different patients. In those consultations, on average (median over GPs), 52.1% (IQR = 49.2% to 59.3%) of patients were female and their median age was 58 years (IQR = 52 to 64). Characteristics of GPs stratified by part-time status are given in Table 1.

**Table 1. GPs (total n = 245).**

| Part-time status | Full-time | 90% | 80% | 70% | 60% | 50% | 40% | 30% |
|---|---|---|---|---|---|---|---|---|
| **Number of GPs** | **68** | **15** | **50** | **22** | **30** | **40** | **15** | **5** |
| Female, % | 12% | 0% | 24% | 41% | 60% | 75% | 80% | 80% |
| GP age groups, % | | | | | | | | |
| 30–39 years | 4% | 13% | 14% | 14% | 23% | 20% | 20% | 0% |
| 40–49 years | 28% | 20% | 34% | 36% | 40% | 32% | 53% | 20% |
| 50–59 years | 43% | 27% | 38% | 27% | 23% | 30% | 20% | 40% |
| 60–69 years | 25% | 40% | 14% | 23% | 13% | 18% | 7% | 40% |
| Self-employed, % (vs. employee) | 84% | 67% | 64% | 41% | 50% | 25% | 47% | 60% |
| In group practice, % (vs. single) | 63% | 87% | 96% | 95% | 97% | 100% | 100% | 100% |
| Urban location, % (vs. non-urban) | 65% | 73% | 78% | 86% | 73% | 85% | 80% | 80% |
| Consultation patient characteristics, median (IQR) | | | | | | | | |
| . . . percent female patients | 50% (48%-53%) | 50% (47%-52%) | 51% (49%-54%) | 51% (48%-56%) | 54% (51%-64%) | 60% (56%-69%) | 59% (52%-64%) | 66% (63%-72%) |
| . . . median age of patients | 61 (56–64) | 63 (54–64) | 57 (51–63) | 60 (55–65) | 56 (48–66) | 54 (48–59) | 49 (45–58) | 53 (49–62) |
| Proportion of workdays worked, median (IQR) total: 246 workdays | 93% (87%-98%) | 99% (85%-100%) | 92% (81%-98%) | 88% (76%-98%) | 86% (73%-96%) | 80% (66%-92%) | 62% (56%-79%) | 75% (71%-87%) |
| Number of individual patients per year, median (IQR) | 1442 (1168–1764) | 1427 (1042–1800) | 1243 (992–1597) | 1088 (828–1401) | 980 (795–1263) | 826 (656–952) | 685 (556–817) | 646 (612–655) |
| **Total consultation counts, median (IQR)** | **6548 (5315–7879)** | **6678 (4876–8029)** | **5740 (4452–7706)** | **4907 (3507–5981)** | **3998 (3110–4217)** | **3151 (2575–3588)** | **1940 (1751–2338)** | **1914 (1563–2214)** |

## Consultation counts per day

The median of the GPs' mean daily consultation counts was 28 (IQR = 22 to 35). Fig 2 depicts the distribution of daily consultation counts stratified by part-time status. Median daily consultation counts increased with part-time status, revealing three steps with similar values (for 40% to 50%, 60% to 70%, and 80% to full-time, respectively).

## Efficiency

GPs' efficiency, based on part-time adjusted total GP consultation counts in 2018, varied with part-time status. Low (30%-50%) part-time workers were less efficient than full-time workers, whereas high (60% to 90%) part-time workers were equal or more efficient than full-time workers (Fig 3).

## Associations of GP- and practice-level factors with total consultation counts

**Crude models.** The GPs' total consultation counts in 2018 were highly associated with part-time status in the crude model. All GPs except for those with 90% part-time status held fewer consultations than GPs working full-time. Furthermore, consultation counts were lower among female GPs (-29.9% consultations), employed GPs (-21.3%), and GPs with high proportions of consultations by female patients (-1.7% per one percent increase in female patients). In contrast, consultation counts were higher among GPs working in single practices

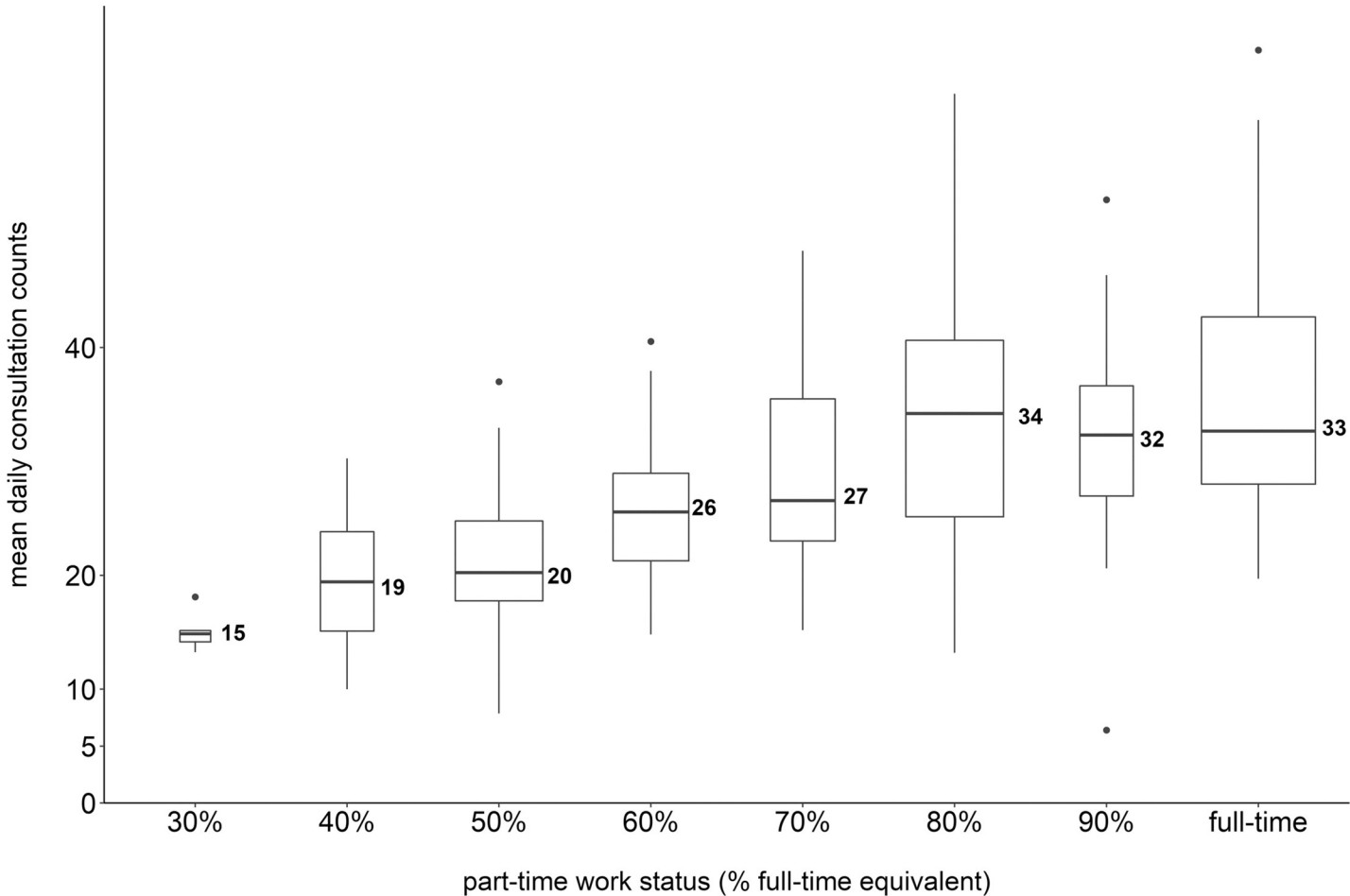

**Fig 2. Mean daily consultation counts.** Boxplots of the GPs' mean daily consultation counts, stratified by part-time status (n = 245 GPs). Widths of boxes are proportional to the square roots of the numbers of GPs in the part-time strata and median values are rounded to whole numbers.

(+37.8%) and GPs with older patients (+1.3% per one year increase in median patient age). The detailed results of the crude analyses are shown in Table 2.

**Adjusted without part-time status.** When adjusting for all variables simultaneously except for part-time status, consultation counts were still lower among female GPs (-19.9%) and higher for GPs working in single practices (+18.8%) and with older patients (+0.7% per one year increase in median patient age, Table 3). Employment status and patient gender were no longer significantly associated with total consultation counts. Instead, the latter were now negatively associated with GP age group 60–69 years (-14.7% with respect to age group 50–59 years).

**Adjusted including part-time status.** Including part-time status in addition to all other predictors dissolved most associations of GP characteristics with total consultation counts. Only oldest GP age (60–69 years) still showed a negative effect (-10.6% with respect to age group 50–59 years), while old patient age showed a positive effect (+0.4% per one year increase in median patient age, Table 3). Part-time status below 90% was still associated with lower consultation counts, but effect sizes were smaller than in the crude model.

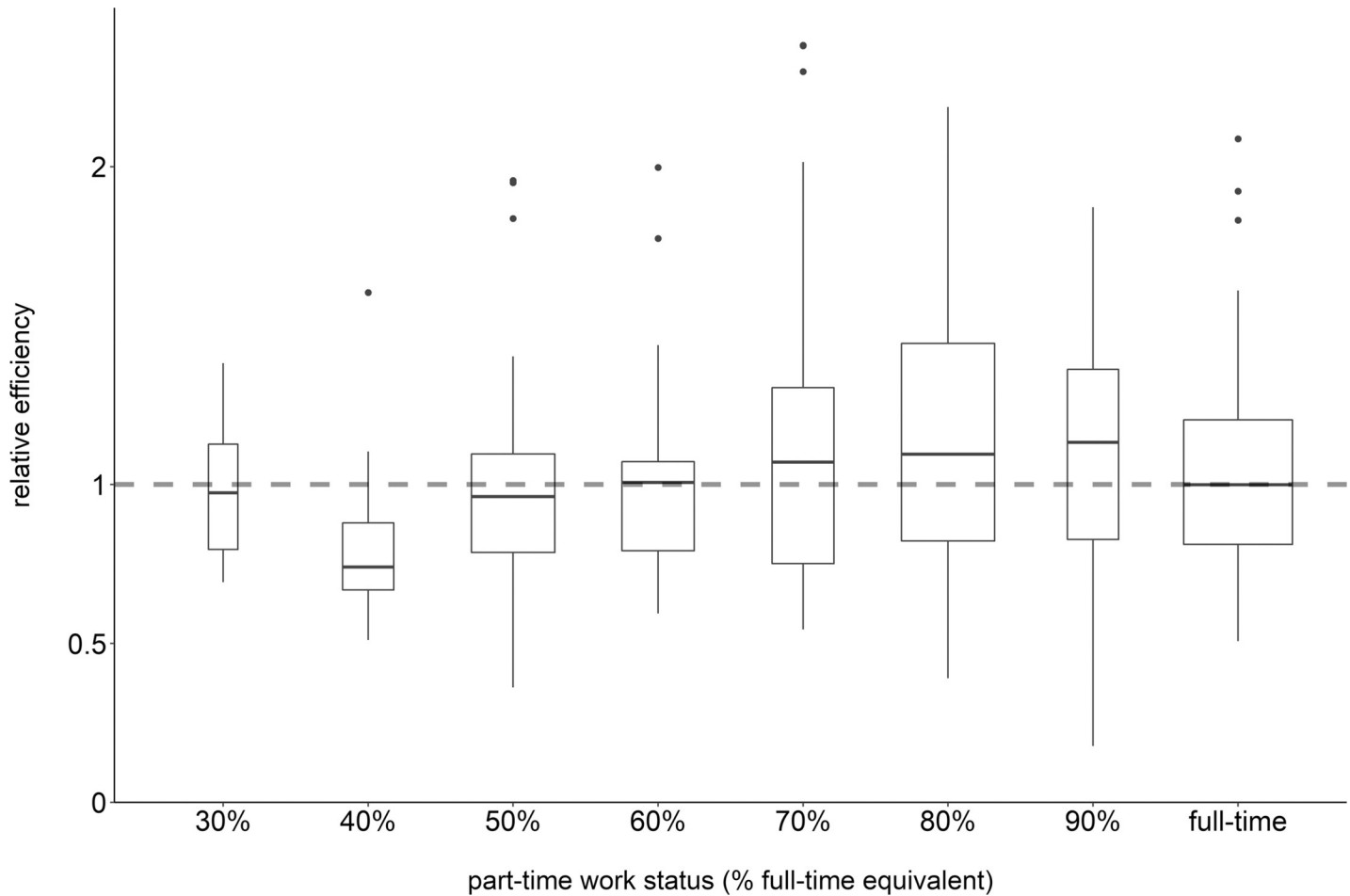

**Fig 3. Relative efficiencies.** Boxplots of relative efficiencies stratified by part-time status (n = 245 GPs). Widths of boxes are proportional to the square roots of the numbers of GPs in the part-time strata. The dashed line represents the reference value (efficiency of full-time workers).

## Discussion

The median of the GPs' mean daily consultation counts was 28 over all GPs, and 33 for full-time workers. Daily consultation counts were non-linearly dependent on part-time status; a plateau was reached at 80% part-time status. High part-time working GPs (60%-90% of full-time) were slightly more efficient than full-time workers, with 90% part-time workers having the same total consultation count as full-time workers. Crude associations alone might suggest that highest consultation counts can be found among male GPs working self-employed in single practices caring for predominantly elderly male patients. However, the multilevel regression models put this into perspective: When adjusting for all variables except for part-time status, the effect of GP gender was reduced, and employment status and patient gender were no longer associated with consultation counts. After additionally including part-time status in the model, apart from part-time status itself, only GP and patient age remained significant predictors of consultation counts.

### Comparison with existing literature

Daily consultation counts as found in our study were similar to results of a European survey conducted in 1993, both over all GPs and adjusted for part-time work [2]. However, for

**Table 2. Crude analyses of total consultation counts.**

| Variable | Change in consultation count (n) | 95% CI | p-value |
|---|---|---|---|
| Part-time status (ref. full-time) | | | |
| *Intercept* | *6924* | *6444 to 7404* | *<0.001* |
| 90% | -331 | -1232 to 570 | 0.471 |
| 80% | -944 | -1537 to -351 | 0.002 |
| 70% | -1516 | -2284 to -749 | <0.001 |
| 60% | -2432 | -3066 to -1797 | <0.001 |
| 50% | -3651 | -4264 to -3038 | <0.001 |
| 40% | -4247 | -5063 to -3431 | <0.001 |
| 30% | -4647 | -5873 to -3420 | <0.001 |
| GP gender (ref. male) | | | |
| *Intercept* | *6260* | *5816 to 6705* | *<0.001* |
| Female | -1873 | -2347 to -1399 | <0.001 |
| GP age group (ref. 50–59 years) | | | |
| *Intercept* | *6009* | *5427 to 6591* | *<0.001* |
| 30–39 years | -879 | -1761 to 3 | 0.051 |
| 40–49 years | -601 | -1288 to 86 | 0.087 |
| 60–69 years | -632 | -1465 to 201 | 0.137 |
| Employment status (ref. self-employed) | | | |
| *Intercept* | *5943* | *5463 to 6424* | *<0.001* |
| Employed | -1265 | -2050 to -480 | 0.002 |
| Practice type (ref. group practice) | | | |
| *Intercept* | *5153* | *4672 to 5634* | *<0.001* |
| Single practice | 1947 | 936 to 2958 | <0.001 |
| Urbanity (ref. urban) | | | |
| *Intercept* | *5309* | *4774 to 5843* | *<0.001* |
| Non-urban | 946 | -9 to 1902 | 0.052 |
| Cons. patient characteristics*: gender | | | |
| *Intercept* | *5564* | *5137 to 5991* | *<0.001* |
| % female | -95 | -122 to -68 | <0.001 |
| Cons. patient characteristics*: age | | | |
| *Intercept* | *5488* | *5043 to 5932* | *<0.001* |
| median age | 73 | 43 to 102 | <0.001 |

Abbreviations: ref. = reference; cons. = consultation; CI = confidence interval

* For continuous predictor variables, coefficients represent the change in consultation count per one unit change.

Switzerland, more recent studies have reported 24–25 consultations per day [1, 22]. Lower consultation counts in Switzerland compared to other European countries are plausible because consultation count has been shown to be inversely related to consultation duration [23], which is known to be longer in Switzerland [24]–arguably due to the Swiss payment system which considers consultation length for remuneration [1]. The discrepancies between our results and figures from previous Swiss studies may be explained by the different types of consultations considered. While previous surveys inquired only face-to-face contacts, we

**Table 3. Multivariable analyses of total consultation counts.**

| Variables | without part-time status | | | including part-time status | | |
|---|---|---|---|---|---|---|
| | Change in consultation count (n) | 95% CI | *p*-value | Change in consultation count (n) | 95% CI | *p*-value |
| *Intercept* | 5994 | 5251 to 6738 | <0.001 | 6734 | 5982 to 7486 | <0.001 |
| Part-time status (ref. full-time) | | | | | | |
| 90% | - | - | - | -87 | -1129 to 954 | 0.869 |
| 80% | - | - | - | -690 | -1362 to -18 | 0.044 |
| 70% | - | - | - | -1249 | -2138 to -360 | 0.006 |
| 60% | - | - | - | -2227 | -3011 to -1444 | <0.001 |
| 50% | - | - | - | -3127 | -3990 to -2263 | <0.001 |
| 40% | - | - | - | -3591 | -4678 to -2504 | <0.001 |
| 30% | - | - | - | -4118 | -5554 to -2681 | <0.001 |
| GP gender (ref. male) female | -1194 | -2016 to -373 | 0.004 | -162 | -879 to 555 | 0.658 |
| GP age group (ref. 50–59 years) | | | | | | |
| 30–39 years | -206 | -1118 to 706 | 0.658 | -308 | -1072 to 455 | 0.428 |
| 40–49 years | -229 | -896 to 438 | 0.501 | -14 | -586 to 558 | 0.961 |
| 60–69 years | -879 | -1640 to -119 | 0.023 | -716 | -1371 to -61 | 0.032 |
| Employment status (ref. self-employed) employed | -594 | -1349 to 161 | 0.123 | -200 | -863 to 463 | 0.554 |
| Practice type (ref. group practice) Single practice | 1128 | 111 to 2144 | 0.030 | 350 | -622 to 1322 | 0.480 |
| Urbanity (ref. urban) Non-urban | 327 | -582 to 1235 | 0.481 | 404 | -437 to 1245 | 0.346 |
| Cons. patient characteristics*: | | | | | | |
| % female | -40 | -83 to 4 | 0.075 | -4 | -42 to 34 | 0.826 |
| median age | 39 | 7 to 70 | 0.015 | 27 | 0 to 54 | 0.049 |

Abbreviations: ref. = reference; cons. = consultation; CI = confidence interval

* For continuous predictor variables, coefficients represent the change in consultation count per one unit change.

considered all types of patient care that led to entries in EMR, including telephone consultations and record reviews.

The association of consultation counts with part-time status was non-linear for both daily and total consultation counts. Daily consultation counts considering only days when the GP actually worked are not representative of the GPs' outputs but rather of their actual working patterns, e.g. full vs. half days. Interestingly, there seemed to be no difference in *daily* consultation counts between 40% and 50% part-time workers, between 60% and 70% part-time workers, and among the above 80% part-time workers, respectively, so the difference in *total* consultation counts must have resulted from the difference in days off (see also Table 1). Total consultation counts in 2018 represent the consultation output irrespective of workday patterns and are thus an appropriate basis for efficiency calculations. We found that high part-time workers (60%-90% of full-time) had a higher efficiency than full-time workers. Though other authors have not stratified part-time status into several categories, they nevertheless observed higher productivity for part-time GPs [25].

As consultation workload was heavily influenced by part-time status, any other variable's association with consultation counts must depend on the variable's own relation with part-time status. Investigations of GP part-time status typically focus on gender differences. In many occidental countries, including Switzerland, female GPs have been found to work part-time, or declare that they plan to do so, more often than male GPs [1, 17, 26–32]. However, many of these studies date back several years and it is hypothesized that work-life-balance choices leading to part-time work might be an issue of ongoing societal change. Therefore, these gender effects could diminish in the future [7, 33, 34]. Today, our crude analysis still revealed a 30% lower crude consultation count for female GPs compared to their male peers. The disappearance of this association after adjustment for part-time working is in line with previous European studies [2, 33]. Interestingly, the association was reduced by one third even in the adjusted model where part-time work was not taken into account, indicating that other variables–such as the GPs' age, practice type, or characteristics of their patient base–co-transmit the effect. Fittingly, our analyses (consistently with the literature [32, 33]) revealed that GPs aged 50–59 years and those who work in single practices–groups where female GPs are underrepresented [2]–held more consultations. Additionally, female GPs have been reported to care for a higher proportion of female and younger patients [2], which was negatively associated with consultation counts in the crude model. Therefore, part of the gender differences in consultation counts can be explained plausibly by different work settings and patient populations.

## Strengths and limitations

To our knowledge, this is the first study of this scale using routine data for a detailed investigation of GP consultation counts. We used a large dataset, containing over one million consultations generated by 245 GPs. The inclusion of part-time status was crucial to give insight into consultation workload of individual GPs, given that part-time work has become increasingly common. The combination of multiple regression models allowed for exploration of direct and indirect effects of the investigated variables on consultation counts.

Our GP sample is representative for the Swiss GP community in terms of gender and part-time status but slightly over-represents younger GPs, GPs working employed and those in group practices in urban and suburban areas [17]. Given that future GPs will tend towards working in such environments, our GP sample may better represent the future workforce. Since using EMR is required for participation in the FIRE project, GPs still operating with paper-based medical records were excluded. This part of the workforce, however, can be expected to become less relevant in the future.

The small number of GPs with very low part-time status caused imprecise estimates, therefore the results within these subgroups should be considered with caution. Further subgrouping of the non-urban GP population in order to e.g. analyze the workload of GPs in rural environments was not possible because the sample sizes were too small. Ultimately, our study disregarded consultations on weekends and public holidays and we expect the true total consultation counts in 2018 to be slightly higher. Excluding weekends and public holidays, however, was necessary because on such days, between-GP as well as within-GP variation of consultation counts was very strong and incompatible with our study aim to model typical consultation counts of Swiss GPs.

## Implications for practice and policy

Knowledge about GPs' consultation counts can contribute to health policy and health economical decisions [7, 14, 35]. Very high consultation counts may be used as indicators for

compromised quality of care and GP work satisfaction, while very low consultation counts might touch on the healthcare systems' financial sustainability and raise concerns about the security of future primary care supply.

Our findings suggest that 60%-90% part-time working GPs are at least as efficient as full-time GPs and that efficiency does neither depend on GP gender, employment status nor practice type. Nevertheless, with part-time work becoming more common, the challenge of recruiting new GPs to secure the future workforce remains.

## Acknowledgments

Our thanks go to Fabio Valeri for statistical support and to the FIRE study group of general practitioners for contributing data to the present study.

## Author Contributions

**Conceptualization:** Yael Rachamin, Rahel Meier, Stefan Markun.

**Data curation:** Rahel Meier.

**Formal analysis:** Yael Rachamin.

**Methodology:** Yael Rachamin, Rahel Meier, Thomas Grischott, Stefan Markun.

**Resources:** Thomas Rosemann.

**Supervision:** Thomas Rosemann, Stefan Markun.

**Visualization:** Yael Rachamin.

**Writing – original draft:** Yael Rachamin, Stefan Markun.

**Writing – review & editing:** Yael Rachamin, Rahel Meier, Thomas Grischott, Stefan Markun.

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
