## [Decision Letter · Decision Letter 0]

21 Nov 2019

PONE-D-19-26151

General practitioners’ consultation counts and associated factors in Swiss primary care – a retrospective observational study

PLOS ONE

Dear Ms. Rachamin,

Thank you for submitting your manuscript to PLOS ONE. After careful consideration, we feel that it has merit but does not fully meet PLOS ONE’s publication criteria as it currently stands. Therefore, we invite you to submit a revised version of the manuscript that addresses the points raised during the review process.

We would appreciate receiving your revised manuscript by Jan 05 2020 11:59PM. To enhance the reproducibility of your results, we recommend that if applicable you deposit your laboratory protocols in protocols.io, where a protocol can be assigned its own identifier (DOI) such that it can be cited independently in the future. For instructions see: http://journals.plos.org/plosone/s/submission-guidelines#loc-laboratory-protocols

We look forward to receiving your revised manuscript.

Kind regards,

Denis Bourgeois

Academic Editor

PLOS ONE

Journal Requirements:

Reviewers' comments:

Reviewer's Responses to Questions

**Comments to the Author**

1. Is the manuscript technically sound, and do the data support the conclusions?

Reviewer #1: Yes

Reviewer #2: Yes

2. Has the statistical analysis been performed appropriately and rigorously? 

Reviewer #1: Yes

Reviewer #2: I Don't Know

3. Have the authors made all data underlying the findings in their manuscript fully available?

Reviewer #1: No

Reviewer #2: Yes

4. Is the manuscript presented in an intelligible fashion and written in standard English?

Reviewer #1: Yes

Reviewer #2: Yes

5. Review Comments to the Author

Reviewer #1: Dear Authors,

thanks a lot for your precious work in this field of primary care where data like this is scare. I do have some minor comments and questions for you to be answered.

Abstract:

line 40: Instead of "not including" I would rather us the wording - "Exclusion of..."

line 46-48: Please consider choosing a more positive formulation rather "not affect.... negatively".

Methods:

line 108: What do you mean by "a spatial typology scheme"?

line 127: Please decide whether you "covariates" or "co-variates" throughout the manuscript.

Results:

line 162-164: Did you correct for half-days in this low part-time workers? 30% might mean 3 half days of work (while the kids are in kindergarten) - which might explain why they only see 50% of the patients a full-time GP sees.

also line 265: Please explain why

table 1 - table 2: How do you explain the difference in median age of patients (73 vs. a maximum of 63 years in 90% part-time workers and even lower in all other groups?

Looking forward to your reply.

Reviewer #2: this study is an interesting and well written study.

Just one remark. In the discussion section: the sub-tittle "summary" should be deleted because it induces a misunderstanding.

6. PLOS authors have the option to publish the peer review history of their article (what does this mean?). If published, this will include your full peer review and any attached files.

Reviewer #1: No

Reviewer #2: No

---

## [Decision Letter · Decision Letter 1]

17 Dec 2019

General practitioners’ consultation counts and associated factors in Swiss primary care – a retrospective observational study

PONE-D-19-26151R1

Dear Dr. Rachamin,

We are pleased to inform you that your manuscript has been judged scientifically suitable for publication and will be formally accepted for publication once it complies with all outstanding technical requirements.

With kind regards,

Denis Bourgeois

Academic Editor

PLOS ONE

Additional Editor Comments (optional):

Reviewers' comments:

Reviewer's Responses to Questions

**Comments to the Author**

1. If the authors have adequately addressed your comments raised in a previous round of review and you feel that this manuscript is now acceptable for publication, you may indicate that here to bypass the “Comments to the Author” section, enter your conflict of interest statement in the “Confidential to Editor” section, and submit your "Accept" recommendation.

Reviewer #1: All comments have been addressed

Reviewer #2: All comments have been addressed

2. Is the manuscript technically sound, and do the data support the conclusions?

Reviewer #1: Yes

Reviewer #2: Yes

3. Has the statistical analysis been performed appropriately and rigorously? 

Reviewer #1: Yes

Reviewer #2: Yes

4. Have the authors made all data underlying the findings in their manuscript fully available?

Reviewer #1: Yes

Reviewer #2: Yes

5. Is the manuscript presented in an intelligible fashion and written in standard English?

Reviewer #1: Yes

Reviewer #2: Yes

6. Review Comments to the Author

Reviewer #1: Dear authors,

thanks a lot for adressing and adapting your manuscript - thus I recommended to accept and publish your work.

Reviewer #2: Thank you for the modifications.

7. PLOS authors have the option to publish the peer review history of their article (what does this mean?). If published, this will include your full peer review and any attached files.

Reviewer #1: No

Reviewer #2: No

---

## [Editor Report · Acceptance letter]

20 Dec 2019

PONE-D-19-26151R1 

General practitioners’ consultation counts and associated factors in Swiss primary care – a retrospective observational study 

Dear Dr. Rachamin:

I am pleased to inform you that your manuscript has been deemed suitable for publication in PLOS ONE. Congratulations! Your manuscript is now with our production department. 

With kind regards,

on behalf of

Professor Denis Bourgeois 

Academic Editor

PLOS ONE